# Sacred Trees, Mystic Caves, Holy Wells: Devotional Titles in Spanish Rural Sanctuaries

Jaime Tatay 

Department of Moral Theology and Praxis of Christian Life, Faculty of Theology, Comillas Pontifical University, 28015 Madrid, Spain; jtatay@comillas.edu

**Abstract:** This paper explores how local, lived religion has creatively linked spiritual insights and popular devotions in ecologically valuable settings helping generate and preserve the rich Spanish biocultural heritage. Focusing on a selection of Sacred Natural Sites (SNS), mostly Marian sanctuaries, it shows that local "geopiety" and religious creativity have generated "devotional titles" related to vegetation types, geomorphological features, water, and celestial bodies. It also argues that, despite mass migration to urban centers, the questioning of "popular religion" after the Second Vatican Council, and the rapid secularization of Spanish society over the past fifty years, a set of distinctive rituals and public expressions of faith—some of them dating back to the Middle Ages—have remained alive or even thrived in certain rural sanctuaries. These vernacular devotions, however, do not necessarily announce the advent of the postsecular. Finally, it suggests that Protected Area (PA) managers, regional governments, custodians, anthropologists, tourism scholars, and theologians should work together in order to analyze, interpret, and help solve the management challenges highly popular SNS face.

**Keywords:** sacred natural sites; Catholicism; lived religion; geopiety; landscape management

## 1. Introduction

Despite the rapid and complex secularization process of Spanish society over the past five decades (Pérez-Agote 2007, 2012; González-Anleo et al. 2021), certain spiritual traditions, pilgrimages, and rituals have remained surprisingly alive or even experienced a significant revitalization (Maldonado 1996; Lois González 2013). Some of these devotional practices date back to the late medieval or the early modern period, others have a more recent origin, and many are linked to Marian hermitages, chapels, shrines or monasteries considered Sacred Natural Sites (SNS) (Wild and McLeod 2008) in or near Protected Areas (PAs) with high ecological value (Tatay-Nieto and Muñoz-Igualada 2019).

Traditional knowledge, values, narratives, beliefs, taboos, and rituals matter from a strictly conservationist point of view (Berkes 2018; Studley 2018; Frascaroli 2013; Colding and Folke 2001). However, as it has been widely discussed since the publication of Lynn White's seminal article, "The Historical Roots of Our Ecologic Crisis" (White 1967), spiritual beliefs, doctrinal systems, and religious practices are a double-edged sword with regard to conservation: they carry values, generate worldviews, and shape attitudes that can either preserve or destroy the natural world (McLeod and Palmer 2015; Bhagwat et al. 2011).

Focusing on a selection of Spanish rural sanctuaries, I argue, in line with Elisabeth A. Allison and John Studley, that, despite the historical shortcomings of spiritual movements and religious institutions, "local people can extend an ethic of care to their biophysical surroundings, through the mediation of personified deities" (Allison 2015, p. 502) and lay participation can play a key role in the ritual protection of SNS (Studley 2018). Since sacred places "have the capacity to also foster attachment, devotion, spiritualy, and a certain 'disposition', 'ethos', and 'worldview'" (Mazumdar and Mazumdar 2004, p. 387), in order to better understand the conservation potential of Spanish SNS, I explore the

symbolic, cultural, and spiritual meaning of sanctuaries placed in natural settings where local, popular devotions have been observed or have even thrived over the past decades.

Surprisingly, despite the profound changes which have occurred in European Catholic countries since Vatican II, some SNS have defied both the secularization of society and the clericalization of religion by means of cultivating a broader understanding of the sacred while generating lively rituals and devotional lay practices rooted in local, folk culture. However, this is not something new; as Per Binde noted, the history of local religion in Catholic Southern Europe "is characterized by a persistent tension between the tendency among practitioners to go beyond the limits of orthodoxy and the wish among the central institutions of the Church to uphold these limits" (2001, p. 21).

This research, thus, resonates with a rapidly expanding body of literature on SNS, but also with the growing academic interest on "lived religion" or "the embodied and enacted forms of spirituality that occur in everyday life" (Ammerman 2014). It also connects with the renewed understanding of "sanctuary" as an appropriate spatial metaphor for the study of contemporary religion (Chappel 2020).

However, when it comes to the current interest in SNS, the spatial turn religion scholars refer to does not necessarily imply a religious revival or the clear evidence of a post-secular trend. Moreover, as Ian Reader has noted, ancient renewed spiritual practices "have not arrested the general decline of church attendance and adherence. Indeed, one might suggest almost the contrary" (Reader 2007, p. 226). In fact, "spaces and objects are imbued with religious meanings in ways that trouble the binaries of public and private, or secular and religious" (Chappel 2020, p. 18). In his extensive review of Spanish sanctuaries, José Miguel Muñoz similarly warned that, "in order not to lose the true meaning of the phenomenon of popular religion, it is important to know that when studying the rural sanctuaries of devotion in Spain, any preconceived idea is broken in the eyes of the researcher because we are close to the ungraspable and intangible, to an unstructured religious world" (2010, p. 447).

Spanish SNS and their surroundings serve today as "multifunctional spaces" (Cànoves et al. 2012) or "servicescapes" (Higgins and Hamilton 2019) where a complex mix of spiritual, cultural, therapeutic, and leisure activities take place (Schnell and Pali 2013). Their nature-related symbolism and the conservation potential of the lively devotional practices that take place in and around these sites remains, however, under-researched. This is certainly the case of highly popular sanctuaries such as *El Rocío* (Andalusia), *Montserrat* (Catalonia), *Guadalupe* (Extremadura) or *Covadonga* (Asturias).

## 2. Research Methods

This study builds on and expands previous research on the conservation potential of Spanish SNS (Tatay-Nieto and Muñoz-Igualada 2019). I systematically searched for bibliographical and internet references to hermitages, shrines, chapels, churches, and monasteries placed in rural settings and wilderness areas. Institutional webpages of these sacred sites, essays on history, ethnology, and anthropology as well as travel guidebooks were thoroughly investigated (Abad León 1990; Aldea 1975; Amengual i Batle 1997; Buxó et al. 1989; Caballero Mújica 1999; Carrasco Terriza 1992; Carreres i Péra 1988; Cebrián Franco 1989; Christian 1976, 1981; Delclaux 1991; Fernández-Ladreda 1989; Fernández Sánchez 1994; Ferri Chulio 2000; Iturrate 2000; Fernández Álvarez 1990; Fuixench 2007; González Echegaray 1993; Lisón Tolosana 1976; López Martín 1998; Llamas 1992; Muñoz 2010; Pérez Ollo 1983; Sánchez Ferrer 1995; Torra de Arana 1996).

In this investigation, which is part of a wider research project, I did not use questionnaires or conduct in-depth interviews. I relied on the (both quantitative and qualitative) analysis of the bibliographical secondary sources and on the dialogue (phone conversations and emails) with members of lay fraternities, custodians and local priests. These provided a total of 574 non-urban, mainly Marian sanctuaries with nature-related names or "natural titles" (Tables 1–4).

**Table 1.** Marian titles related to plants.

|  | Title | n. |
| --- | --- | --- |
| Rose | *Roser, Rosa* | 64 |
| Holly oak | *Encina, Encinar, Encinillas, Carrasco, Carrascal, Lluc* | 38 |
| Common hawthorn | *Espino, Espinar, Arantzazu* | 30 |
| Field elm | *Olmo, Olmeda, Olmacedo, Oms, Omedes* | 24 |
| Olive | *Oliva, Olivar, Olivares* | 22 |
| Pine | *Pino, Pinar, Pinarejo* | 22 |
| Common grape vine | *Vid, Viña, Viñedo, Parral, Vinyet, Raïmat* | 20 |
| Elmleaf blackberry | *Zarza, Zarzuela, Zarzaquemada, Navalazarza* | 18 |
| Rosemary | *Romero, Romeral* | 9 |
| Willow | *Salcedo, Salceda, Salcedón, Sargar, Saz* | 8 |
| Reed | *Juncal, Junquera, Junqueres, Xunqueira* | 7 |
| Heath | *Brezo, Brezales, Bruguers* | 6 |

**Table 2.** Marian titles related to geomorphological features.

|  | Title | n. |
| --- | --- | --- |
| Rock/Stone | *Peña, Berrocal, Muskilda, Peñitas, Piedraescrita, Peñas Albas, Peñarrota, Peñahora, Roca, Rocamayor, Saliente, Llosar* | 20 |
| Mountain | *Montaña, Monte, Montemayor, Montesclaros, Montserrat, Mont, Sierra, Monfragüe, Mont, Monsacro, Monsalud, Toro, Moncayo, Bellmunt* | 17 |
| Valley | *Valle, Val, Roncesvalles, Vallivana, Valmayor, Valvanera, Valdesalce, Navahonda, Miravalles* | 13 |
| Meadow | *Vega, Soto* | 10 |
| Cave | *Cueva, Cueva Santa, Covadonga, Balma, Cova, Cuevita* | 8 |
| Hill | *Pueyo, Puy, Puig, Puig-graciós, Tura* | 6 |
| Field | *Campo* | 5 |
| Way | *Camino* | 4 |
| Plain | *Llano, Llana, Llanos* | 4 |
| Mountain Pass | *Puerto, Colls* | 3 |
| Ravine | *Hoz, Tallat* | 3 |
| Canyon | *Angosto* | 2 |
| Wilderness | *Yermo* | 2 |
| Vulcano | *Volcanes* | 1 |

**Table 3.** Marian titles related to light and celestial bodies.

|  | Title | n. |
| --- | --- | --- |
| Light | *Luz* | 8 |
| Star | *Estrella* | 3 |
| Moon | *Luna* | 1 |
| Sun | *Sonsoles* | 1 |

**Table 4.** Marian titles related to water.

|  | Title | n. |
|---|---|---|
| Fountain | *Fuente, Fuentesanta, Fuencisla, Valdefuentes, Caldas, Aguas Santas, Aguas Vivas, Hontanares* | 13 |
| Snow | *Nieves* | 8 |
| River | *Río* | 2 |
| Sea | *Mar* | 2 |
| Swamp | *Tremedal* | 1 |
| Lake | *Lago* | 1 |
| Seport | *Puerto* | 1 |

A selection of the most popular ones was inventoried with the objective of assessing their spiritual meanings, ritualized traditions, and associated devotional practices. Finally, the religious significance and the theological underpinnings of the most salient nature-related devotions in these sanctuaries were debated with colleagues. The insights gained from the research were also discussed with peers in order to achieve a greater understanding of the interaction between popular piety, local devotions, natural titles, and the conservation potential of rural sanctuaries. Finally, I focused on three highly popular SNS set in PAs—*El Rocío*, *Covadonga*, and *Montserrat*—to illuminate the role some devotions and rituals may have played in their conservation status and religious vitality.

This interdisciplinary research on Spanish SNS, thus, stands at the intersection of several disciplines: anthropology, history, cultural geography, tourism studies, conservation biology, and religious studies. Here, I discuss only some of these intersections. Space does not permit to take up all of them, but they can form the subject of future studies.

### 3. Local Religion and Marian Sanctuaries in Natural Settings

Spain is the third tourist destination in the world (United Nations World Tourism Organization 2018). Around 83.7 million people visited the country in 2019, setting an all-time record. Although the exact numbers are difficult to estimate, pre-COVID-19 counts by custodians and regional governments show that the most-popular SNS in Spain attracted millions of visitors every year. The sanctuary of El Rocío (4 m), which sits right next to Doñana NP, the sanctuary of Montserrat (3 m) in the Parc Natural de la Muntanya de Montserrat, and the sanctuary of Covadonga (1.5 m) at the entrance of Picos de Europa NP stand among the favorite destinations. The main pilgrimage to El Rocío may draw up to 1 million devotees. Moreover, every weekend, thousands of pilgrims, hikers, climbers, and tourists intermingle by the Benedictine Monastery of Montserrat and near the grotto of Covadonga. All three sacred sites share in common being very active Marian spiritual centers in PAs of high ecological value.

The cult of Mary has been described as "baptised paganism" (Benko 2004, p. 4; McKenna 2011). It was developed by Justin Martyr (100–165 CE) and, since the 4th century, became deeply intertwined with the goddess cult (Artemis, Diana, Isis, Cybel). Mary eventually became Mother of God, ever-virgin, intercessor, mediatrix and Queen of Heaven (Benko 2004; Baring and Cashford 1991). She was refashioned as mother goddess and Magna Mater (as Black Madonna) and associated with fertility, health, earth worship (earth, water, mountains, rocks), and the sacred feminine (Christian 1981). Many Cybel sites were dedicated to Magna Mater and became synonymous with venerating the earth (Benko 2004, pp. 213–14). The development of the Mary cult also involved the "construction of a sacred Marian topography" (Benvenuti 2017, p. 49). Water, fountains, rocks, mountains, caves, and sites previously dedicated to spirits/goddesses became Marian sanctuaries (Curchin 2014, pp. 158–59; Benko 2004, p. 213).

In Spain, animism continued during the Visigothic kingdom and, in the northwest, Celtic cults persisted for many centuries as folk practices, which in some cases were

Marianized (Curchin 2014). In the south and the Mediterranean coast, Roman cults were transformed into those of the Saints and the Virgin. All over the peninsula, the Roman Catholic Church raised churches, hermitages and sanctuaries at traditional cultic sites such as springs, rivers, caves, and cliffs where locals paid tribute to pagan gods. In the 13th century, probably influenced by the Cistercian reform, Marian shrines began to proliferate in Spain and overshadowed previous patron saints (Figure 1). Through a complex historical process, popular devotion first universalized the Marian cult—thus, simplifying the diversity of the cult to the saints—but soon after re-localized it again while generating hundreds of distinctive titles or "advocations" (from the Latin *advocatio*, pleading the cause of another). Many of these advocations, such as *Rocío* (Dew), *Montserrat* (Rugged Mountain), *Espino* (Common hawthorn), *Roser* (Rose), *Peña* (Rock), *Fuensanta* (Holy Well), and *Covadonga* (or *Cova-donga*, Cave of the Lady), are nature related.

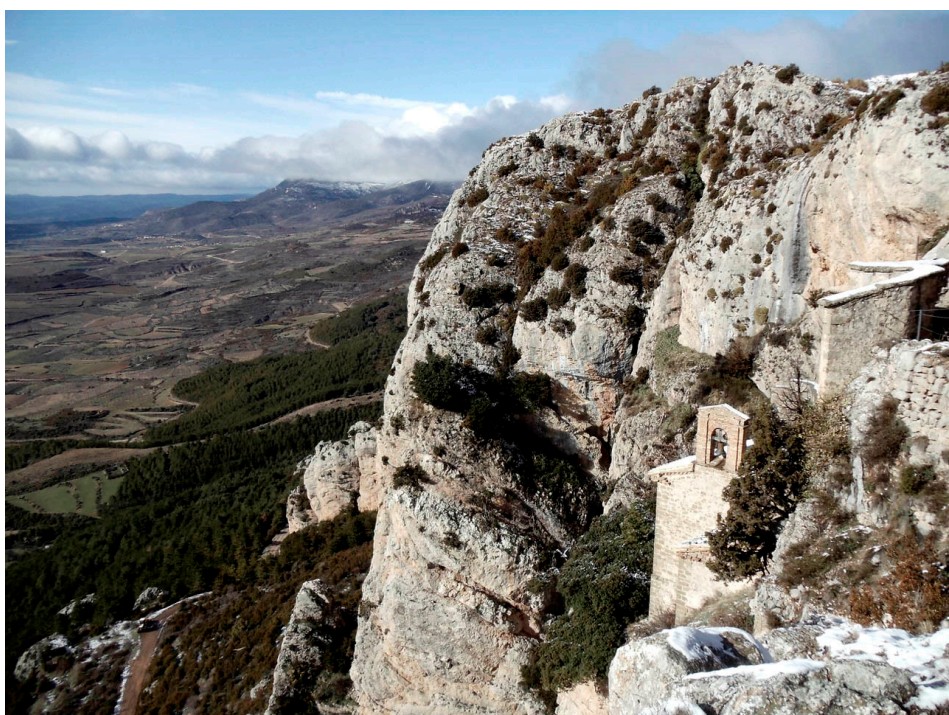

**Figure 1.** The 13th century *Virgen de la Peña* (Virgin of the Rock) chapel and hermitage in Aniés, Aragón (Photo: Jaime Tatay).

Later on, from the 16th through the 18th centuries, ecclesiastical authorities, likely influenced by the Golden Legend (Hani 1995), put in writing the history of these enclaves. They became sacred or "enspirited" (Studley 2018, 2019) sites because spiritual-like events occurred there. In most of the accounts, the unexpected discovery of the site by a shepherd and the stories of miracles performed by Mary play a prominent role. As time went by, an increase in the number of wonders and healings helped make the local image popular. When it comes to the origins of Marian apparitions in the wilderness, a distinctive narrative structure has been identified in several studies (Binde 2001).

Joan Prat, for example, concludes his ethnographic analysis of Catalan sanctuaries affirming that "a shepherd, and less frequently a shepherdess, observes the atypical behavior of an ox or bull of his herd, which by roaring or digging, insistently calls attention to its owner or guardian. The latter, intrigued or surprised, begins a systematic search near the *source*, *cave*, *rock*, *tree* or *thorny bush* where the inspired animal drives him, and ends up discovering the image of a Virgin" (1989, p. 221).

Like Prat, Frabizio Frascaroli (2016) identified a similar structure in the foundation stories of most "natural shrines" in Central Italy and underlined the symbolic importance of mobile pastoralism in their emergence. In Spain, these shrines also represent a type of

"spiritual settlement" of the mountain regions (Garganté and Solà 2017). From a ritualistic perspective, anthropologist Alfredo Rodríguez identified another narrative pattern in his analysis of the miraculous power of Spanish Marian sanctuaries: "The *sanctuary* is configured as one of the three elements of a certain type of cult, the other two would be the *image* that is venerated and the *miracles* that are performed, with a basic scheme that is repeated: the Virgin appears to someone, usually a shepherd, in an unpopulated area, and [later on the shepherd] discovers an image, hidden from time immemorial [...] After a series of vicissitudes, which often include a certain amount of incredulity on the part of the neighbors and local authorities, a sanctuary is erected at the place of the apparition, and then the miracles that increase its fame take place" (Rodríguez 2000, pp. 162–63).

In sum, just as the *tree* (thorny bush)—*rock* (cave)—*water* (fountain) are some of the key natural elements that articulate the sacredness of a SNS (Figure 2), the *sanctuary—image—miracle* also configure a complex unity mediated by distinctive, local narratives, devotions, and rituals. According to José María Fuixench, "Water, tree, and rock are elements that have always formed a universal cosmos, wherever a hermit dwells. A trilogy of life where the water symbolized purification; the tree, regeneration; and the rock, immortality" (2007, p. 17).

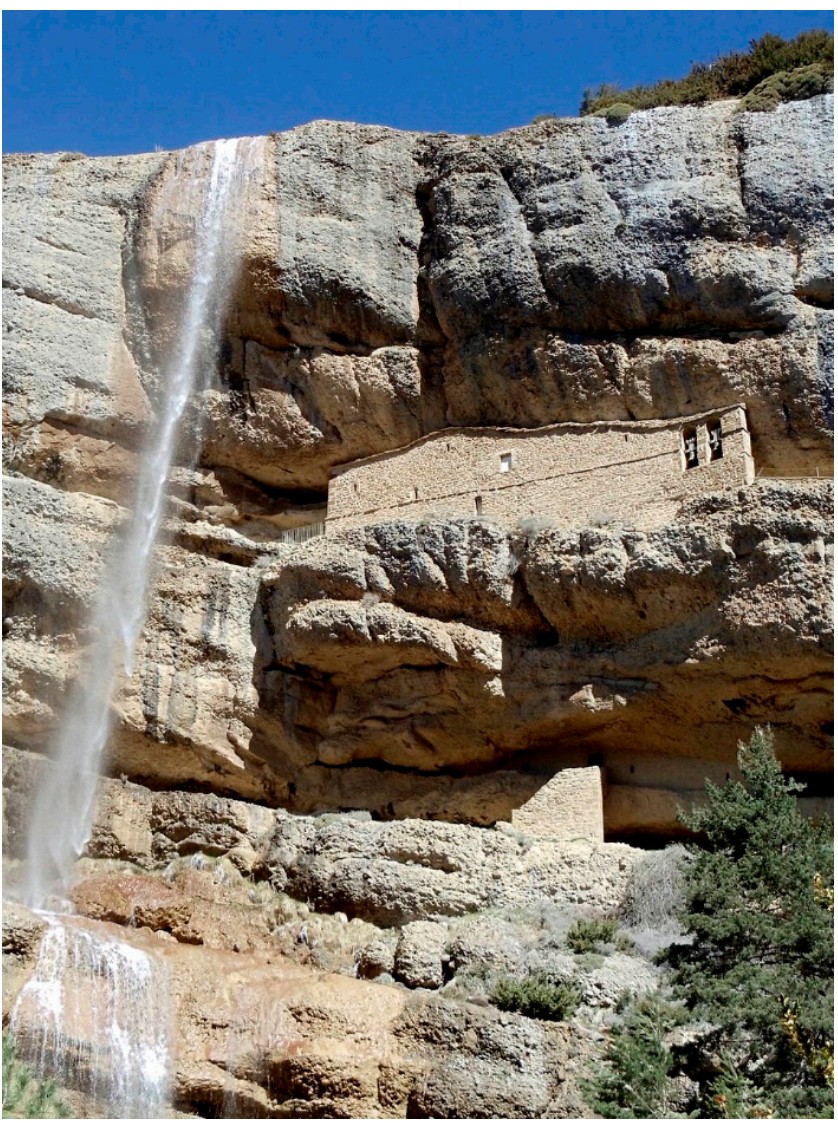

**Figure 2.** The Yebra de Basa Hermitic complex near *Santa Orosia*, Aragón. (Photo: Jaime Tatay).

During the yearly celebrations, these isolated, liminal places become intense centers of social interaction. The historical events that gave birth to the sanctuary are remembered and devotees engage in a "contractual relation" (Arregui 1989; Coggins and Hutchinson 2006) and a ritualized "symbolic exchange" (Prat 1989) that involves candles, food, gifts, and votive offerings such as flowers. Historian of religion Mircea Eliade (1968) noted that, whatever the complexity of a religious feast, it is always a sacred event that took place *ab origine* and must be ritually reenacted to be kept alive. For Prat, during the festivities Marian images undergo a "process of symbolic condensation" that both humanizes Mary and transmits her heavenly power to heal and transfer certain supernatural virtues to humans. Local narratives, rituals, and devotions are deeply connected to these symbolic exchanges.

## 4. Pilgrimage, Religious Tourism, Ecotourism, and Secularisation

The powerful attraction of SNS in the secular, Post-Vatican-II era is an intriguing characteristic of the contemporary Spanish religious situation. Nevertheless, we should not rush and be misled by the growing numbers of visitors before drawing conclusions. Tourism scholars have warned over the past three decades that sacred sites and pilgrimage routes are a holy ground and we need to tread carefully.

Valine L. Smith already noted in the 1990s that, "because of the increased secularization of religion in the West, distinctions between pilgrim and tourist are quite diffuse" (Smith 1992, p. 15). Two decades later, María Cruz Porcal (2006) acknowledged the difficulty of drawing a line between pilgrims and tourists in the famous *Javieradas* pilgrimage to the sanctuary of Saint Francis Xavier (Navarra). More recently, Gemma Cànoves and her team showed in their study of Spanish sacred places that "religious tourism and cultural tourism form a joint trend that is an expression of the commercialization of culture or, more exactly, of religion and popular devotion" (2012, p. 283).

However, this is not just a Western phenomenon. In her comparative study of traditional pilgrimage and modern tourism in the Indian Himalayas, Shalini Singh concluded that, in both religious and secular contexts, "pilgrimages in recent times are fuzzily interpreted [ . . . ] making it almost impossible to draw a clear distinction between true pilgrims and mainstream tourists", and a new, emerging type of "environmental pilgrims" (Singh 2005, pp. 221–23). In a similar way, other scholars have more recently argued, in line with the pilgrim-tourist continuum model proposed by Smith, that "spiritual tourism" (Lopez et al. 2017) and "ecotourism" (Yoder 2017) can be interpreted as forms of religious tourism that lie in between (sacred) pilgrimage and (secular) tourism.

Furthermore, from a religious, ritualistic perspective, "setting apart" is central to the practice of sacred-making. This is why, in line with Mircea Eliade (1968) and Roy Rappaport (1971), Siv Ellen Kraft (2011) and Josep Maria Mallarach (2018) have suggested that the legal establishment of PAs that preserve ("set apart") certain landscapes based on their ecological value can be interpreted as a substitute for ritual and taboo in secular contexts. Legalization, thus, may well have become a substitute for sacralization.

In sum, many different types of visitors (pilgrims, religious tourists, secular tourists, ecotourist, and environmental pilgrims) are being attracted to SNS, although for different reasons (Mantsinen 2020). Contemporary sacred sites around the world have, thus, become complex multifunctional spaces (Higgins and Hamilton 2019), social meeting points, places of rest, secret "regional capitals" (Díez Taboada 1989), therapeutic centers, religious supermarkets (Cànoves et al. 2012), and settings where the spiritual meets the numinous.

## 5. Geopiety, Nature-Related Devotional Titles, and the Spiritual Significance of Nature

We are all embedded in a network of physical, psychological, and emotional relationships from the moment we are born—not just human relationships, but also relationships with the landscape and the surrounding nature. Local people often exhibit very strong "place attachment" (Mazumdar and Mazumdar 2004) to the landscape and, in some cultures, even engage in the "nurture" of topographic features to appease the numina who

inhabit them (Studley 2019). "The cultural and spiritual significance of nature has been defined as the spiritual, cultural, inspirational, aesthetic, historic and social meanings, values, feelings, ideas and associations that natural features and nature in general have for past, present and future generations of people—both individuals and groups. The attributes of nature conveying such significance range from species of flora and fauna to natural features to entire landscapes and waterscapes" (Verschuuren et al. 2021, p. xiv).

Japanese cultural historian Tetsuro Watsuji (1961) argued that our inner landscape somehow correlates and is shaped by the outer, physical landscape. Chinese American humanistic geographer Yi-Fu Tuan also noted that peoples and places are intrinsically bonded. He referred to this universal relationship as "topophilia" (Tuan 1961), only a few years before Erich Fromm first used the word "biophilia" (Fromm 1964). Shortly after, another American cultural geographer, John Kirtland Wright, coined the term "geopiety" (Wright 1966, pp. 251–52). The neologism expresses the worship of, and reverence for certain elements or features of the natural landscape. Wright understood it as a type of religious awareness concerning manifestations of geodiversity.

Following Tuan and Wright, other authors have later suggested that religion is directly influenced by the landscape (Gualteri 1983) or could even be considered a geographical phenomenon. For instance, J. Gary Knowles stressed the practical, action-oriented attachment and reverence for particular places in the environment: "geopiety connotes action and notions of responsiveness towards a place that is regarded as sacred" (Knowles 1992, p. 9). From a sociological perspective, in an effort to analyse behaviour and belief as a cultural whole, Bronislaw Szerszynski (1997) distinguished four types of *ecological piety*, interpreted as "an orientation to the sacrality of nature embodied in people's lives" (1997, p. 50). Environmental psychologists have also emphasized that "ties to the sacred provide people with an identification with place which may persist through time and across generations" (Mazumdar and Mazumdar 2004, p. 395). More recently, Singh argued that "the geographical notion of *genus loci* has been employed to exemplify the fundamental quest for 'geopiety' attained through the unification of the pilgrim's intrinsic belief with its external location" (2005, p. 215).

Yet, this universal quest is more strongly related to certain religions. For Kevin Kiernan, natural landforms, including mountains, rocks, caves, and islands, are regarded as sacred sites in many cultures and "figure prominently in traditional and polytheistic faiths and residually in monotheistic faiths" (Kiernan 2015, p. 177). However, although Christianity explicitly rejects polytheism, "animistic beliefs and images of communion with nature" form an integral part of Catholic "popular religious practices" (Binde 2001, p. 15). Several authors have argued that, as the Marian cult developed in the Iberian Peninsula, it absorbed animism—the Celtic nature spirits and the mother goddess cult—and vestiges remain to this day (Curchin 2014; McKenna 2011). The many natural titles and nature-related devotional practices found across Catholic regions could be regarded, thus, as animistic expressions of popular, local geopiety.

Juan José Cebrián Cebrián Franco (1989, pp. 301–5) has identified four types of Marian titles in Spain: derived from toponyms, related to traits of the images, focused on details of the miraculous apparition, and linked to aspects of human liberation. For Muñoz (2010, p. 45), these devotional titles are either historical, toponymical, architectural, geographical or floristic. Previous research has shown that forests, trees, shrubs, and flowers played a prominent role in articulating the many plant titles or "verdant advocations" (Tatay-Nieto and Muñoz-Igualada 2019) of Spanish SNS. In fact, out of the 574 nature-related rural chapels, hermitages, shrines, and monasteries analyzed in this study, a significant 73.5% (422) have a vegetation or plant title. The remaining 26.5% (140) of the sanctuaries are named after a geomorphological feature (100), a water-related element (29), a celestial body (15) or—very rarely (Figure 3)—an animal (4). I now turn to analyze the meaning of the most common natural titles.

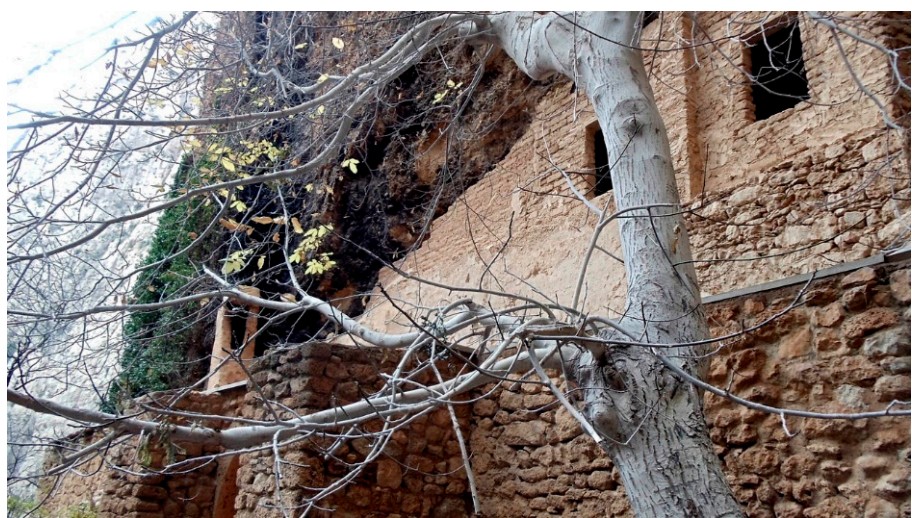

**Figure 3.** *San Martín de la Val de Onsera* (Saint Martin of the Bear Valley) is the only SNS named after an animal in the Aragonese Pre-Pyrenees (Photo: Jaime Tatay).

### 5.1. Sacred Groves, Church Forests, Holy Trees

Sacred trees and forests represent a long-held tradition across cultures and religions. "There can be no question of the forests as a consecrated place of oracular disclosures; as a place of strange or monstrous or enchanting epiphanies; as the imaginary site of lyric nostalgias and erotic errancy; as a natural sanctuary" (Harrison 1992, p. 121). This is probably why the conservation potential of Indian sacred groves (Bhagwat et al. 2005), Ethiopian church forests (Klepeis et al. 2016), Tibetan old growth trees (Salick et al. 2007) and sacred groves in many cultures around the world is attracting growing academic interest.

In Spain, as in many other industrialized European countries, "tree worship has for the most part disappeared" (Crews 2003, p. 43) and there is not a strong, sacred forest tradition anymore. However, the reverence for trees or "arborphilia" (Greenberg 2015) is still alive in certain settings. Groves and gardens around hermitages, chapels, cemeteries, monasteries, and sanctuaries still share some of the sacredness of the holy sites (Stara et al. 2015; Frascaroli 2013; Vaitkevičius 2009). Forests, springs, and iconic landforms are key natural elements in most Spanish SNS. A few species of trees and shrubs such as roses (*Mare de Déu del Roser*), holly oaks (*Virgen de la Encina*), common hawthorns (*Nuestra Señora del Espino*), elms (*Virgen del Olmo*), pines (*Virgen del Pino*), olive trees (*Nuestra Señora del Olivo*), and grape vines (*Virgen de la Vid*) stand out as the most common plant-titles (Table 1).

In some local devotions, the flower itself becomes central. For instance, at the *Mare de Déu dels Lliris* (Mother of God of the Lilies) sanctuary in Alcoy (Alicante), in the *Parc Natural del Carrascar de la Font Roja*, one of the local custodians noted that, "according to tradition, the image of the virgin miraculously appeared on 21 August, in the middle of summer, when lilies do not usually bloom, among the gorse. When the lily bulb was peeled, the image of the virgin appeared. A second priest peeled another lily, and the image appeared again. Every year, in pilgrimage people go in procession to the oak forest from the village". One of the monks at Montserrat also mentioned the importance trees and plants play in the *Virolai*, the hymn of the Virgin of Montserrat. The hymn begins with this verse: "*Rosa d'abril, morena de la serra, de Montserrat estel* [ . . . ] *Roser del Cel; Cedre gentil, del Líban sou corona, Arbre d'encens. Palmera de Sió...*" (Rose of April, Brunette of the Mountain, Star of Montserrat [ . . . ] Rose Tree of Heaven; Gentle cedar, Crown of Lebanon, Tree of Incense, Palm tree of Sion . . . ).

However, why do so many popular devotions and titles relate Mary to the shrubs and the trees? Several authors have shown that there are many similarities between Artemis—the fertility goddess of Asia Minor, spirit of forests, and deity of the wilderness—and the Virgin Mary (Harrison 1992; Mallarach 2013; Ionescu 2016). In Greek antiquity,

the virgin goddess Artemis was "the noumenal spirit of the forests which gives birth to a multiplicity of species (forms) that preserve their originary kinship within the forest' network of material interdependence" (Harrison 1992, p. 29). The cult of Artemis declined in the 4th century, during the reign of Theodosius, and her temple (in Ephesus) was closed in the early 5th century. This coincided with the veneration of Mary as *Theotokos* (Mother or God) which was sanctioned at the Council of Ephesus in 431 CE. Similar parallels have been found across Latin America (Damian 1995); for instance, between the Aztec goddess of fertility, sacred trees, and the Virgin (Granziera 2012).

In Christian theology, Mary is not a person of the Holy Trinity, yet intense popular devotion and veneration gave rise to her cult and fostered an impressive iconographic development of her image. In most processions, her images are surrounded by flowers and often the final stretch of the route is carpeted with branches, leaves, and petals (Figure 4).

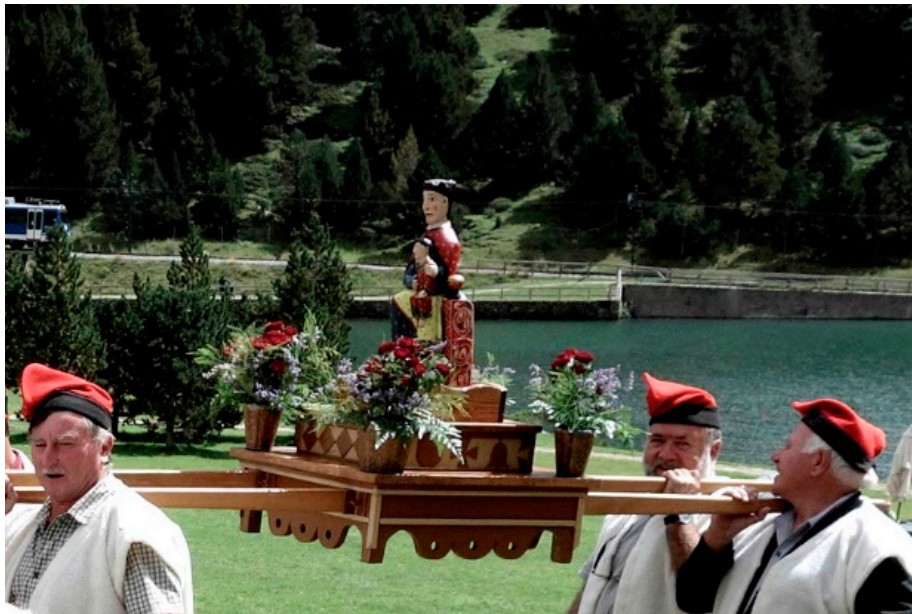

**Figure 4.** The patroness of the Pyrenees, *la Mare de Déu de Núria*, is carried to her sanctuary on 1 September 2016, the feast of St. Giles, patron saint of shepherds (Photo: Josep Maria Mallarach).

It is then plausible to argue that early devotion to Mary was influenced by other cults and found inspiration in the universal tree-of-life motif, setting the Virgin's apparitions in forests, "channeling the life-giving force of trees through the use of arborescent titles" (Tatay-Nieto and Muñoz-Igualada 2019, p. 3). According to Carole M. Cusack, holy groves also represent the site of sacred and social rituals, "the stability of the cosmos and of society" (Cusack 2011, p. 172). The tree, and its derivative, the pillar (also a very popular Spanish Marian devotional title, *La Virgen del Pilar*), are symbolic indicators of the *axis mundi*, the center of the world.

The abundance of plant titles in Marian SNS and the scarcity of "animal titles"—eagle (*Nuestra Señora del Águila*), goat (*Santa María de Cabrera*), bull (*Mare de Déu del Tura*), and bee (*Virgen de las Abejas*), are some of the few exceptions—suggests that Mary may also represent, following Hegel's juxtaposition, the innocence, quietness, and tranquility of "flower religions" versus the aggressivity, anxiety, and sacrificial dynamic of "animal religions". Maybe because plants, like Mary, "embody the kind of detachment human beings dream of in their own transcendent aspiration to the other, Beauty, or divinity" (Marder 2013, p. 12).

Yet not only Catholic devotions place the Mother of God on top of trees or hidden in thorny bushes. Vegetation plays a prominent role in ancient and current devotional practices and rituals as well. In Spanish, those who engage in a *romería*—a short or medium annual pilgrimage to a local sanctuary—are called *romeros* (literally "rosemary holders", from the woody, perennial bush with fragrant evergreen leaves *Rosmarinus officinalis*). Petals, leaves, and branches of different types of plants also blanket the floor ahead of

the procession that leads the devotees to the sanctuary (Figure 5). Additionally, of course, bouquets of flowers are usually offered as a sign of gratitude and left at the altar or around the image presiding the sacred site.

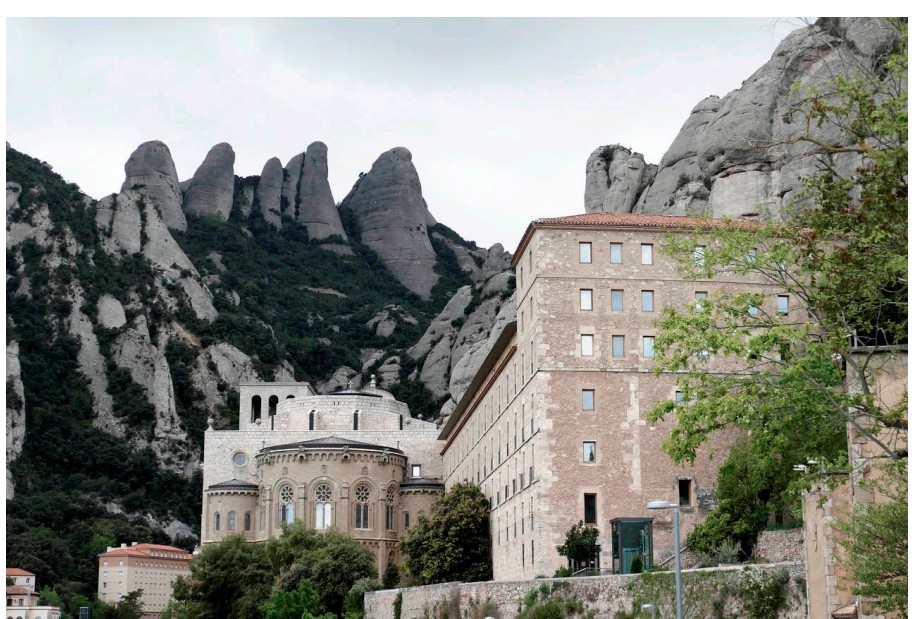

**Figure 5.** The impressive monastic complex of Montserrat is surrounded by mountains and forests (Photo: Pixabay).

### 5.2. Sacred Mountains, Mystic Caves, Holy Rocks

Across times, cultures, and religions, geographical features and geomorphological forces have implications for the ways native peoples understand themselves and make meaning in relation to their landscape. Most Spanish rural sanctuaries are placed in iconic settings that offer a wonderful panoramic view of the countryside. It is no wonder that "one of the most constant impressions that visitors have when they go to Marian shrines is that they are all—or almost all—located and built in the most privileged places that the surrounding nature offers" (Prat 1989, p. 223). Similarly, Muñoz has argued that, "in the location of the Spanish rural sanctuaries, some clear geographical clichés converge (the mountain, the sickle, the river, the fountain, the island, the peninsula, the singular tree, the hanging stones, the forms of wind origin, etc.)" (2010, p. 537).

Although Christians do not revere natural forms, devotional titles related to landforms and geological features are not uncommon, especially within the Eastern Orthodox Church. They express a type of local, affective connection to particular traits of the landscape, a sort of indigenous geopiety towards the (Christianized) *genus loci*. In Spain, rocks (*Peña*, *Roca*, *Berrocal*, *Muskilda*), valleys (*Valle*, *Val*), mountains (*Montaña*, *Monte*, *Mont*, *Monsacro*, *Monsalud*, *Toro*, *Moncayo*, *Bellmunt*, *Sierra*), meadows (*Vega*, *Soto*), caves (*Santa Cova de Montserrat*, *Covadonga*, *Cueva*, *Balma*), hills (*Pueyo*, *Puy*, *Puig*), fields (*Campo*), plains (*Llano*), paths (*Camino*), and mountain passes (*Puerto*, *Coll*) stand out among the most popular geomorphological and landscape Marian titles (Table 2).

Devotional titles related to the sun (*Nuestra Señora de Sonsoles*), the moon (*La Virgen de la Luna*), the stars (*La Virgen de la Estrella*), and light (*Nuestra Señora de la Luz*) are much less frequent, but still can be found in rural sanctuaries (Table 3).

For thousands of years, religions around the world have used caves as spaces for protection, meditation, art, and burials. As Verschuuren et al. affirm, "many caves were used already in pre-Christian times as dwellings, burials, worship sites, and shelters for mobile pastoralism (transhumance). After Christianization, they have been revered as hermitages and sites of divine apparitions" (2021, p. 62). This has been certainly the case in Spain, where some very important SNS are placed in caves (Figure 6). As one of

the custodians at the *Santuario de Covadonga* affirmed, "the hierophanic elements of the mountain, the cave, and the water play an important role in the (local) religiosity. The image of the Virgin of Covadonga is venerated *inside* the cave". Likewise, at the *Virgen de la Cueva Santa* (Virgin of the Holy Cave) sanctuary in Altura (Castellón), the local priest underlined the importence of *Cueva* (cave) as an advocation or Marian title and brought to the fore the fact that she was the Patroness of Speleologists.

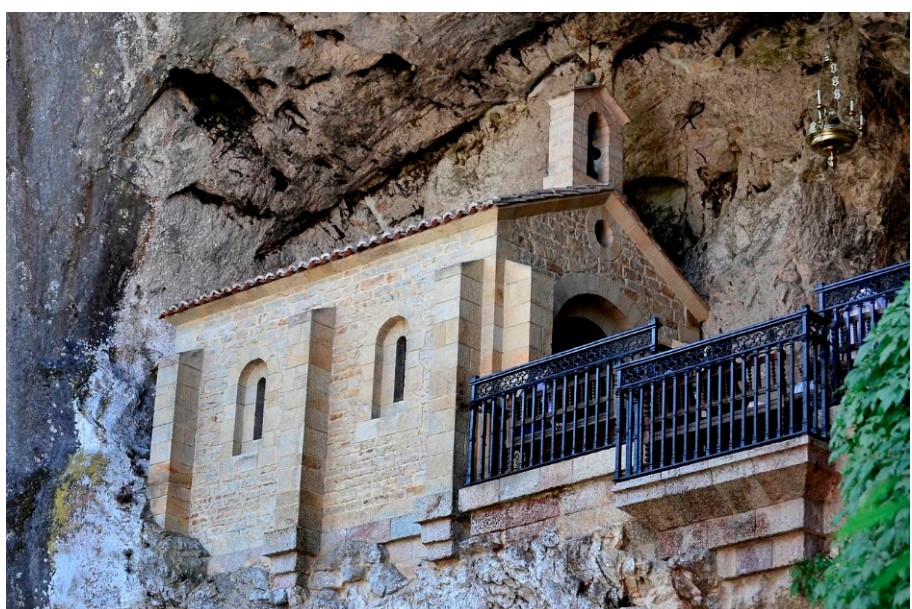

**Figure 6.** The chapel of Covadonga (Photo: Lourdes Álvarez).

In a recent study on the Cave-Sanctuary of *Cueva Santa del Cabriel* (Central-Eastern Spain), Sonia Machause and her team found that, from the Late Chalcolithic to the present day, the ritual use of the cave has only been interrupted during two periods. Additionally, even then, "it is tempting to think that some notion of sanctity persisted among the people living in the surrounding areas" (Machause et al. 2019, p. 72). They also argue, in line with previous research, that "the physical characteristics of the cave [ . . . ] impart an intrinsic sensory power to the site, helping to create a strong ritual atmosphere" (p. 62).

Across pilgrimage routes and SNS throughout the world, rocks and pebbles (Figure 7) have ritual significance (Sneath 2006). Heaps of small stones and cairns are commonly found at many sacred wells. In the hermitage of *San Andrés de Teixido* (Saint Andrew of the Yew) on the Northern Galician coast, *romeros* used to throw stones at particular sites forming *humilladeros* or *milladoiros*—large mounds or piles of stones—near the sanctuary or at particular crossroads (Figure 8). This ancient, pre-Roman practice acquired penitential meaning during the Christian era.

After a period of decline, stone piling is coming back among contemporary pilgrims around the world, reaching the religious and agnostics alike. In a recent study conducted in Western Ireland on modern and medieval pilgrimage practices, Ryan Lash suggests that, whereas in medieval times "pilgrims' contribution to the formation of the complexes by depositing quartz pebbles would have enhanced this sense of shared participation in ascetic devotion", today, through "embodied encounters with stones", contemporary pilgrims are generating new "enchantments" (Lash 2018, p. 297).

In sum, rocks, trees, and water are easily "enspirited" (Studley 2018) and become settings for ritual practices. They are key elements that articulate the sacredness of SNS. In fact, their symbolism is closely related.

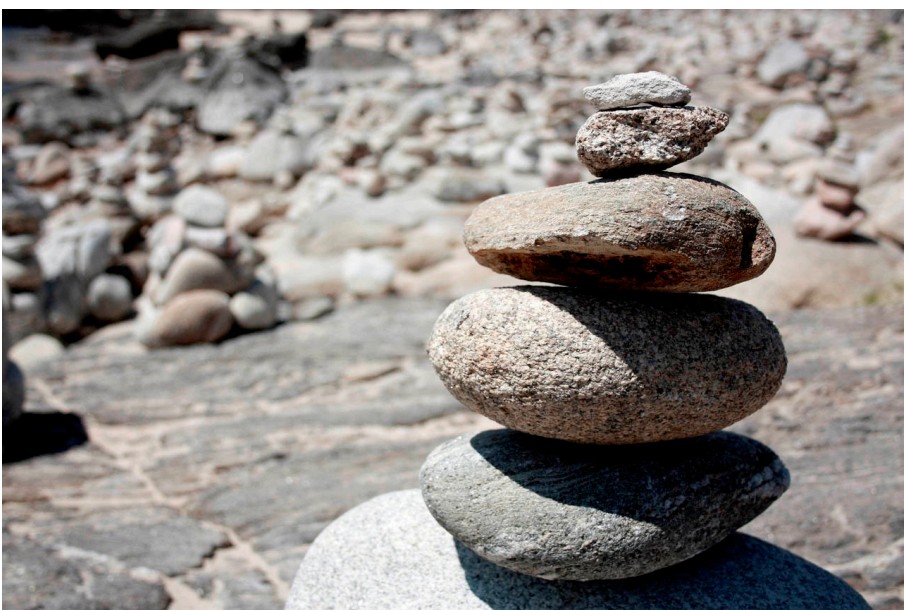

**Figure 7.** Stone heaps along the Camino de Santiago (Photo: Pixabay).

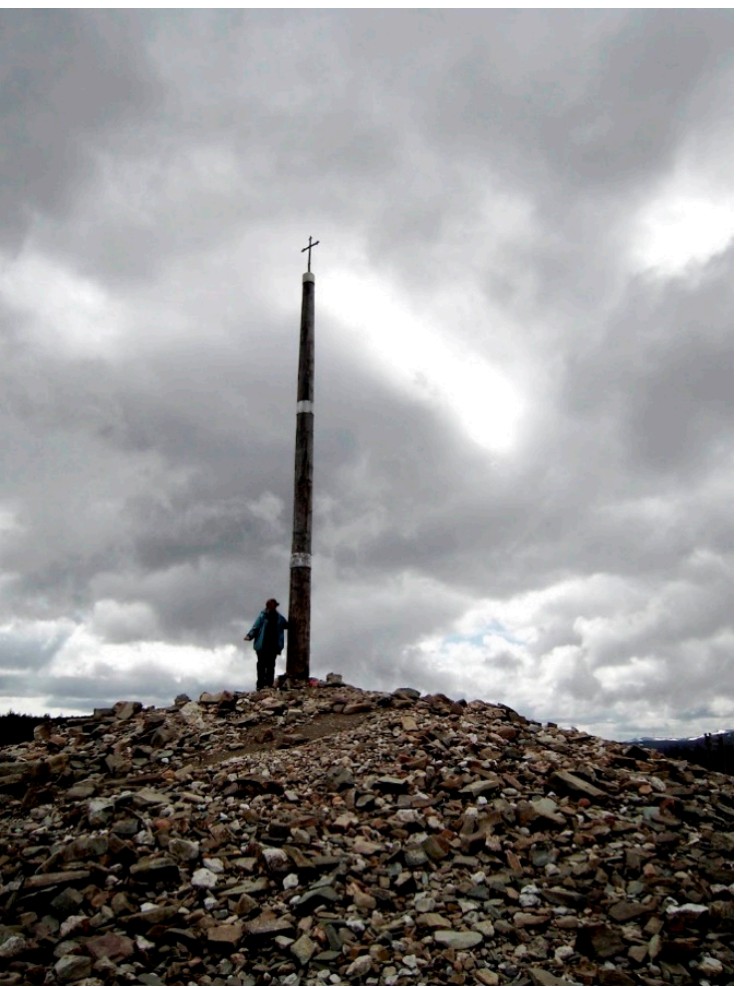

**Figure 8.** Over the years, pilgrims along the *Camino de Santiago* have left thousands of stones and pebbles forming large *humilladeros*, like the one sitting under *La Cruz de Ferro* (Photo: Alejandro Coma).

*5.3. Holy Wells, Sacred Fountains, Mystic Dew*

The existence of holy wells and sacred fountains is a global phenomenon that spans geographies, cultures, and religions. As one of the leading experts on the topic, Gary R. Varner, puts it: "Sacred wells have been with humankind for thousands of years. Their symbolism of wisdom, holiness, and healing and divine feminine power mixes with the darker aspects of life and death, the underworld and the underworld's denizens. There are thousands of holy wells, streams, rivers and lakes throughout the world, in some locations undoubtedly still secret" (Varner 2009, p. 151).

Ronan Foley has also noted that "well settings are often framed by streams, stones and trees, which act as markers of a connection between the grounded earth and the sky" (Foley 2010, p. 470). In Spain, most holy wells and sacred fountains sit right by a chapel or a hermitage and are surrounded by trees. The life-giving character of fountains makes them "sites of powerful sacred and symbolic meanings for local communities" (Allison 2015). Like in Tronchón (Teruel), where the custodian of the *Santuario de la Virgen del Tremedal* (literally, Virgin of the Swampy Terrain), said –using the present tense– that "the virgin appears on top of a rock and works three miracles: she gives the one-armed shepherd back his arm, offers a treasure for the construction of the hermitage, and makes water spring from the rock".

In Christian countries, "over several hundred years, mysterious ladies in white and the Virgin Mary have been reported at or near many holy wells" (Varner 2009, p. 145). Certainly, this has been the case all over Spain (Table 4). Water-related Marian titles are common and refer to fountains (*Aguas vivas, Fuencisla, Hontanares*), holy springs (*Fuensanta, Aguas Santas*), hot springs (*Caldas*), dew (*Rocío*), swamp (*Tremedal*), or snow (*Nieves*). There are also a few sacred sites named after a river (*Río*), sea (*Mar*), lake (*Lago*), and seaport (*Puerto*).

The stories, rituals, and devotions developed around holy springs are usually bound to the therapeutic power of the water. Lourdes, in Southern France, stands out as a paradigmatic case of this connection. Although, in other less well-known sites, the link is also present. In his study of Spanish rural sanctuaries, José Ignacio De Arana has noted: "An important group of these [devotions] are related to [ . . . ] certain places that already possessed healing virtues for pagan peoples. For thousands of years, and in all cultures, some places on earth enjoyed mysterious powers to alleviate all the ailments of the body or some in particular [ . . . ] But above all, these are springs whose water is capable, either ingested or applied externally, of modifying our organism by making its ills disappear" (De Arana Amurrio 1997, p. 586).

The many stories, rituals, and traditions developed around the highly popular Spanish Marian title *Virgen de la Salud* (Our Lady of Health) connect local devotions to water and healing. However, there are other, non-therapeutic uses of holy water in SNS. For example, in Covadonga the waters of the torrents that pass through the cave are collected in the *Fuente del Matrimonio* (Marriage Fountain), a goblet-shaped fountain in which, according to tradition, young women who drink from it will soon be married (Figure 9). In Montserrat, one of the many devotional titles of the Virgin, "*Mística Font de l'Aigua de la Vida*" (Mystical Source of the Water of Life), also expresses the close connection between water and Mary in the popular imagination.

Paradoxically, the only sacred site named after dew (*Virgen del Rocío*), one of the rarest Marian natural titles, became the most popular of all Spanish SNS (Figure 10). The sanctuary is built "in a chosen place, at the edge of the marshes and at the edge of the current of water called *Madre de las Marismas* (Mother of the Marshes)" (Aldea 1975, p. 2234). Another title of this same Virgin is *Reina de las Marismas* (Queen of the Marshes).

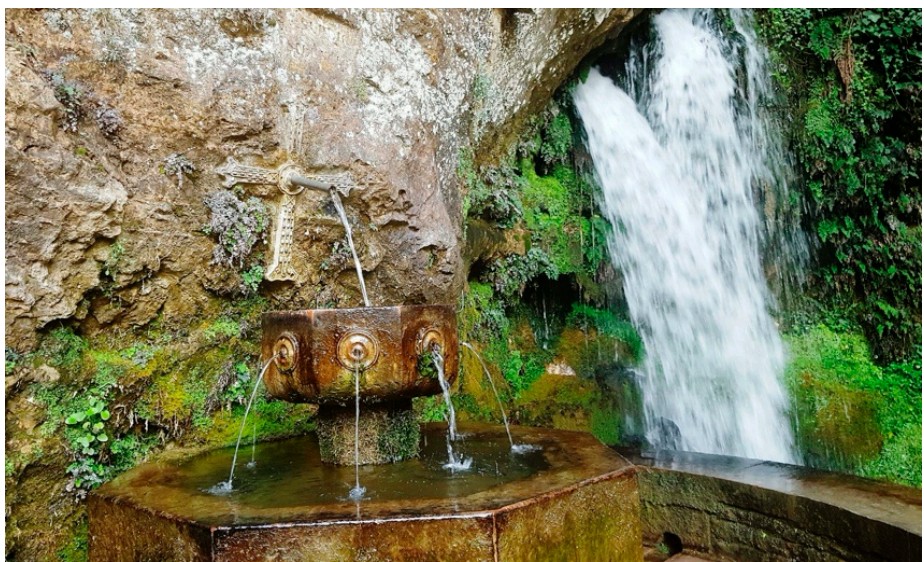

**Figure 9.** *Fuente del Matrimonio* (Photo: Archivo del Santuario de Covadonga).

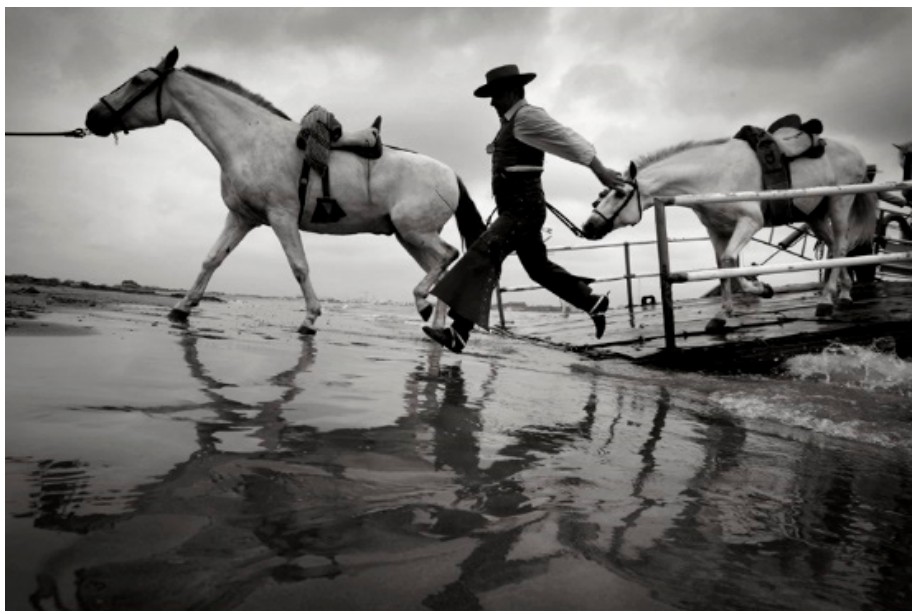

**Figure 10.** Pilgrims on their way to *El Rocío* (Photo: José Luís Roca).

In the baroque mentality, holy water from heaven was a divine remedy for drought. This is the reason why, in a plenary agreement of the town of Almonte held in 1726, the city hall decided to ask its Patroness to bring the *"Santo Rocío de sus aguas" (The Holy Dew of its waters)*. Nowadays, even though modern irrigation systems pump underground water during dry years, devotion to the *Madre de las Marismas* has not declined. On the contrary, it has increased substantially over the last decades, attracting ever larger numbers of pilgrims, since, as Antonio Ariño wrote three decades ago, "old rituals with a clear agrarian functionality (a plea for rain) are taken up again in a different socioeconomic context and are transformed into symbols of a real or longed-for identity" (Ariño 1989, p. 483).

## 6. Discussion and Conclusions

Since there is already evidence that the most visited Spanish rural sanctuaries are placed in or near PAs of high ecological value (Tatay-Nieto and Muñoz-Igualada 2019) and have been established and maintained by local communities for centuries, if not millennia,

it seems plausible to argue that folk traditions, popular devotions, and religious rituals have functioned as conduits for collective (environmental) ethical behavior. Eugene N. Anderson, an American cultural ecologist, concluded from his research on indigenous peoples in several continents that environmental management requires both a strong, simple ethical code and some form of religion that involves people emotionally (1996, p. 162). The question that remains, however, is whether custodians and local communities could actively promote ecological awareness and work along with environmental managers in order to protect these sites in an age of mass tourism and ecological decline.

Unfortunately, recent research on the relationship between SNS, pilgrimage and tourism shows that "there is a disconnect between religious views of the environment and the actions of pilgrims and religious tourists, who, because of their views of the sacred nature of their travels, blame other tourists and travelers for the environmental ills at sacred sites" (Olsen 2020, p. 33). However, Olsen also argues that, when faith leaders emphasize the ecological impacts of spiritual journeys, pilgrims and religious tourists become more environmentally sensitive. Therefore, what can we conclude after analyzing the many natural titles and some of the lively devotional practices that take place in Spanish rural sanctuaries?

First, given the global trend to interpret religious beliefs, narratives, and practices in the light of contemporary environmental concerns and the high ecological value of SNS and their surroundings, these sanctuaries are being increasingly perceived by the conservation community as key biocultural repositories that enable both the conservation of biodiversity and the survival of local forms of knowledge, traditions, and rituals. There is, thus, room to affirm that religion serves as a "natural vehicle for carrying messages about conservation and wise management" (Anderson 1996, p. 82).

Second, SNS, even when they become primarily tourist destinations, are never completely secular. They still posses a *spiritus loci* and undergo a sacralisation process that links the environment, history and local identity. "Attachment to the sacred, forged through visits, rituals, stories, and artifacts [ . . . ] can link a diverse and dispersed community of believers in a collective bond" (Mazumdar and Mazumdar 2004, p. 395). The many nature-related titles described in this study as well as the narratives, rituals, and devotional practices that take place in these rural sanctuaries not only express "place attachment", but also a type of local "geopiety" or "topophilia". Since SNS and beautiful landscapes are increasingly considered "natural amenities" and "spiritual resources" (Ferguson and Tamburello 2015) used by the population to (re)connect with the sacred, they could prove valuable venues for possible partnerships between PAs managers, custodians, and local communities.

Third, if "setting apart" is central to the practice of sacred-making, then the legal establishment of PAs can be interpreted as a substitute for ritual and taboo in our secular age. We may also wonder whether judicial processes are replacing ancient rituals leading to a different type of re-sacralization outside the control of both institutional religion and political authorities. This may well have been the case in highly popular Spanish SNS that have witnessed an unprecedented increase in the number of visitors since they were declared PAs.

Fourth, many sacred sites are under increasing pressure around the world due to a variety of reasons, including the commercialization of culture and the mutation of religious beliefs. In Spain, the rise in mass tourism and the revitalization of certain spiritual practices, rituals, and devotions in some SNS after the Second Vatican Council represents a complex challenge to managers and custodians alike. This renewed interest is, thus, a double-edged sword that could generate a sense of sacredness, "geopiety", and respect towards the natural world or further deteriorate these fragile enclaves. There is already evidence that Spanish SNS have promoted conservation. Yet, care needs to be taken to actively preserve these sites based on their religious and cultural symbolism. Otherwise, the increase in visitors may harm their surrounding environments.

Fifth, the pre-COVID-19 growth of pilgrimage and the interest in visiting sacred sites in Spain is a complex and ambiguous phenomenon that may reflect, despite its spiritual appearance, a secular trend. Pilgrims, religious tourists, cultural tourists, and ecotourists flock to certain SNS en masse, but perceive them in different ways. Since the role of sustainable tourism in supporting conservation is becoming increasingly important, PA managers, custodians, regional governments, anthropologists, cultural geographers, tourism scholars, and theologians should work together in order to analyze, interpret, and help solve the management challenges highly popular SNS face.

This study is not free from limitations. It only comprises a preliminary examination of some Spanish SNS. For a more comprehensive understanding of the interaction between rural sanctuaries, local religion, tourism, and conservation, there is a need to research other places and conduct in-depth interviews with PAs managers, custodians, and visitors.

**Funding:** This research received funding from the Universidad Pontificia Comillas research project: "Sitios naturales sagrados y espacios naturales protegidos: explorando experiencias de los actores e interacciones entre dos lógicas de conservación de la Naturaleza a través del estudio del caso del Monasterio del Paular en el P.N. de la Sierra del Guadarrama".

**Informed Consent Statement:** Informed consent was obtained from all subjects involved in the study.

**Acknowledgments:** The author thanks Liza Zogib, Salvador Ryan, Amparo Merino, and Josep-Maria Mallarach for providing constructive comments, which improved the article.

**Conflicts of Interest:** The author declares no conflict of interest.

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
