# Peer review of "Sacred Trees, Mystic Caves, Holy Wells: Devotional Titles in Spanish Rural Sanctuaries"

_religions, doi:10.3390/rel12030183_

Round 1
Reviewer 1 Report
This is an interesting study on an important subject, but the reader winds up learning more about other SNS rather than those studied by the author. I feel like more attention could be given to the author's own methods and findings, whereas now, more attention seems to be given to contextualizing the findings within other scholarship on SNS and conservation. The original research is the part of this paper that will be of greatest value to other scholars of SNS, so it should get a bit more space in the paper. This would just entail expanding the methods and findings a bit.
Second, I am not a fan of starting sections with long quotes that are then disconnected from the rest of the section (i.e., the start of sections 1, 3 and 4). Instead of leaving a long quote hanging, integrate it into your actual text. Otherwise, it isn't clear why the quote is there and it doesn't specifically support the arguments.
Here are some other points to consider/address:
Second paragraph: SNS and PA should be spelled out on the first use in the paper (not just the abstract).
Line 39: typo, should be "double-edged." As well, needs more detail. They are a double edged sword with regard to conservation.
Line 87 says devotions were debated with custodians. So you did interviews? If so, describe them a bit more, how many did you do? What did people say?
Line 98: does this mean it is the third most popular tourist area?
Line 177: cut "as I already pointed out."
In the end, we learn that many of the Marian shrines have connections to natural features, and that these connections might related to older religious traditions. But do these connections actually promote conservation in these specific places, or is the argument just that they can be used to promote protection? If it is the second option, that's not a very strong argument, but it would still need to be articulated more directly. Maybe something like, care needs to be taken to promote protection of these sites due to their religious connections, otherwise, the increased visitors might love the areas to death and wind up harming the local ecosystems. Or something like that. In the end, the final conclusion can be stated more strongly.
Author Response
Thank you for your time and interest in this paper.
I expanded the research methods and tried to state more clearly the originality/contribution of the paper.
Long quotes have been included in the text. Thank you also for the little suggestions/corrections, they improved the manuscript.
No, we didn't conduct in-depth interviews. Since this piece is part of a broader research project, we plan to use semi-structured interviews in the future. I mainly relied on secondary sources, phone conversations and emails. I did also stated this clearly in the methodology chapter.
There is already evidence that Spanish SNS have promoted conservation (I stated this more clearly), but the question remains whether these sites could promote environmental literacy/awaraness in an age of mass tourism. I tried to address this to in a chapter on tourism-pilgrimage and in the conclusions, which have been expanded.
Thank you again for your review.
Reviewer 2 Report
Review – Spanish SNS
Line 48/49 Sacred/SNS
Sacred needs to be defined – I prefer enspirited (See Studley 2019: xx)
SNS needs to be categorized – Perhaps you could use ontic and epistemic (ontic for enspirited and epistemic for imagined or visualised sites such as mandalized landscapes (Huber 1999)
Other than contractual relations what are the behavioural expectations (environmental protection) in a Marian SNS?
Do you need to say more about the legal or governance status of Marian SNS
Line 110 Marian Cult
Much more required about its growth and development and the implications for enspirited SNS- See below
Line 121 “Become holy” (as a result of a “spiritual event”)
Holy lacks spiritual substance as it just means consecrated – surely enspirited would be a better term (Allison 2009)
Line 125 Apparitions
More required in terms of the nature of the other-than-human persons that appear – are they numina and if so do they adopt the persona of the Virgin Mary. What are they trying to achieve in terms of their status when they communicate with human persons (Christian 1981, von Daniken 1975: 72, 213)
Line 138 Marian Sanctuaries
More needed about their history – were they goddess or pagan sites before – are they enspirited (or made holy) as a result of ritual activity? Are they protected out of deference to the resident numina or only based on so-called “geopiety”. Do relational epistemologies play a part in sanctuaries between humans and other-than-humans?
Other than contractual relations what are the behavioural expectations (environmental protection) in Marian Sanctuaries?
Do you need to say more about the legal or governance status of Marian Sanctuaries
Line 155 Contractual relations
This is an important comment and is a common feature of communities who engage in “contractual reciprocity” with the numina who reside in SNS (Coggins and Hutchinson 2006)
Line 160/266 Channels/Channeling
You might need to use another word as readers might assume you are referring to spiritism or meditation
Line 196 topophilia and geopiety
Are these just convenient exogenous expressions for “outsiders” to use when observing the behaviour of local people who they don't fully comprehend?. In my research (30 yrs plus) local people engage in the “nurture” of topographic features to honour or appease the numina who inhabit them. They do exhibit very strong place attachment to topographic features (such as SNS) but that is not the same as geopiety. None of the people I have researched has ever used expressions such as topophilia or geopiety or their equivalents (in Tibetan, Chinese or Nepalese)
Line 211 natural landforms are regarded as SNS and figure in “polytheistic and monotheistic faiths”
Most of the worlds SNS (especially outside formal Protected Areas) are in the homelands of animistic people (See Studley and Bleisch 2018). What about animism in Spain? What happened to the Celtic nature spirits and the mother goddess cult that existed in Spain before the development of the Mary cult (See Curchin 2014: 158,159 McKenna 2011)?. As the Mary cult developed it absorbed animism and vestiges remain to this day. It is perhaps more evident in South America or the Philippines but it still exists in Spain. Amy Whitehead (2018) argues that touching Marian shrines is animistic and Trexler (1991) argues that madonnas are animistic. The “contractual relations” or “ritualised symbolic exchange” you refer to (Line 155) allow other-than-human persons to enspirit the statue, cave, mountain, stone, forest or tree and you correctly state that ritual exchange “must be reenacted to be kept alive” (Line 159) and (more importantly) for the resident numina to legally remain.
Line 235 Old-growth Tibet trees
source required
Line 237 324 Worship
I know you don't use the term worship but you quote it twice
I think venerate is a better term. Some Roman Catholics claim they don't worship Mary they venerate her.
Line 250 (and 280) “Similarities between Artemis and the Virgin Mary”?
This only tells half the story!
There is evidence that the Virgin Mary replaced Artemis and adopted her role as Queen of Heaven, fertility goddess and spirit of forests. The cult of Artemis declined in the 4th century (during the reign of Theodosius) and her temple (in Ephesus) was closed in the early part of the 5th century. This coincided with the veneration of Mary as Theotok (Mother of God) which was sanctioned at the Council of Ephesus in AD 431. The devotees of Artemis transferred their devotion to Mary.
Line 338 Heaps of small stones and cairns
These are common throughout the world where they often have ritual significance. Some stones are enspirited (Hallowell 1960) and cairns are often the residence of local numen (Sneath 2006).
Line 357 Enchanted
This term lacks spiritual substance and enspirited might be better (Allison 2011)
Mary Cult
The author needs to provide more background to the Mary cult because most non-Roman Catholics have no idea there is a major difference between the Mary (the mother of Jesus) and the pantheon of spirits/goddesses at the root of the Mary cult.
The latter is not the same Mary that appears in the gospels who plays a subordinate role to her son and as a human woman found favour with God. She is a model of loving obedience to something higher than herself (Baring & Cashford 1991)
The Mary cult has been described as “baptised” paganism (Benko 2004: 4, McKenna). It was developed by Justin Martyr (AD 100-165) and since the 4th Century became an outgrowth of the goddess cult (Artimus/Diana/Isis/Cybel) and Mary became:- “mother of god”, an “ever-virgin”, who was immaculately conceived and assumed (into heaven) and also became an Intercessor, mediatrix and Queen of Heaven (Hall 2009: 18, Benko 2004: 4,14,15, 218, Baring and Cashford 1991). She was refashioned as “mother goddess” and “Magna Mater” (as Black Madonna) and associated with fertility, health, earth worship (earth, water, mountains, rocks) and the sacred feminine (Hall 2009: 18,21,82 Vanderford 1945: 73-76, Benko 2004: 15, 213,214, Christian 1981: 21). Many Cybel sites were dedicated to Black Madonna (Benko 2004: 213,214) and according to Alfonso el Sabio venerating Mary became synonymous with venerating the earth (Vanderford 1945: 73).
Importantly for this study, the development of the Mary cult also involved the Marianization of Topography i.e. water, fountains, rocks, mountains, caves (Benvenuti 2017: 49) and often on sites previously dedicated to spirits/goddesses (Curchin 2014: 158, 159 Denning & Phillips 1986: 147, Benko 2004: 213, Wacks 2017)
The cult was used for the absorption of Muslims/Jews/local deities in Spain, IPs in South America and for superstitions connected with some SNS (Hall 2009: 10, 18,82 Benvenuti 2017: 47).
There is no doubt ( Curchin 2014: 158, 159) that animism was widespread in Celtic Spain where it was manifest in mother-goddess and nature cults (mountain and woodland spirits and water nymphs or fairies). Animism continued during the Visigoth kingdom where some of the local female deities include Mari, Ataecina, Nabia and the Nymphae (Curchin 2014: 158, 159, Wacks 2017, Wikipedia 2020, McKenna 2011) and a more extensive list exists (See McKenna 2011). In northern Spain, Celtic cults persisted well into the twentieth century as folk practices which in some cases were Marianized. In the south and the Mediterranean coast, Roman cults were transformed into those of the Saints and the Virgin. All over the peninsula, the Roman Catholic Church raised churches, hermitages and sanctuaries at traditional cultic sites such as springs, rivers, caves, and cliffs where locals paid tribute to pagan gods. Celebrations of solstices and other events marking the agricultural cycle were likewise covered with a veneer of Catholic doctrine but were essentially pagan in substance and symbology (Wacks 2016)
In terms of South America, which is a good example The cult was invoked to incorporate other religions into a Marian framework by “intercultural mingling”. (Hall 2009: 81, 82, 151 Brewer 2003: 10,11). The Spanish used the cult to absorb the local goddesses (Pachamama) and Mary is represented in paintings as a rock/mountain and is considered a numina or huaca (Hall 2009: 140,141,142, Damian 50,55)
References
Allison (2009) Enspirited Places, Material Traces: The Sanctified and the Sacrificed in Modernizing Bhutan. PhD. University of California-Berkley, California.
Baring, A. and Cashford, J. (1991) The Myth of the Goddess: Evolution of an Image. Penguin Books
Benko, S. (2004) The Virgin Goddess: Studies in the Pagan and Christian Roots of Mariology. Brill
Benvenuit, A. (2017) ‘Shrines and Holy Places in Tuscany between Pilgrimage and Religious Tourism: The Difficult Relationship between Knowledge and Valorization’. Almatourism 16, 39–50
Brewer, M. (2003) From Myth to Reality: Performing the Devil and Pachamama in the Carnival of Humahuaca. MA. University of Chicago
Christian, W. (1981) Apparitions in Late Medieval Spain. Princeton University Press
Coggins, C. and Hutchinson, T. (2006) ‘The Political Ecology of Geopiety: Nature Conservation in Tibetan Communities of Northwest Yunnan’. Asian Geographer 25 (1–2), 85–107
Curchin, L. (2014) Roman Spain: Conquest and Assimilation, Taylor and Francis
Damian, C. (1995) The Virgin and the Andes: Art and Ritual in Colonial Cuzco. Greenfield Press
von Daniken, E. (1975) Miracles of the Gods. Souvenir Press
Denning, M. and Phillips, O. (1986) The Magical Philosophy; Book 3 - The Sword and the Serpent. Llewellyn Publications
Hall, L. (2009) Mary, Mother and Warrior: The Virgin in Spain and the Americas. University of Texas Press
Hallowell, A.I. (1960) ‘Ojibwa Ontology, Behavior and World View’’. in Culture in History: Essays in Honor of Paul Radin. New York: Columbia . ed. by Diamond, S. New York: Columbia University Press, 1–25
Huber, T. (1999) The Cult of Pure Crystal Mountain: Popular Pilgrimage and Visionary Landscape in Southeast Tibet. vol. a. Oxford: Oxford University Press
McKenna, S. (2011) Paganism and Pagan Survivals in Spain up to the Fall of the Visigothic Kingdon, Theophania Publishing
Sneath, D. (2006) ‘Ritual Idioms and Spatial Orders: Comparing The Rites for Mongolian and Tibetan “Local Deities”’. in The Mongolia-Tibet Interface. ed. by Bulag, U. and Diemberger, H. Leiden: Brill, 135–158
Trexel (1991) Public Life in Renaissance Florence, Cornell University Press
Vanderford, K. (1945) Alfonso El Sabio: Setenario (in Spanish). Instituto de Filologia
Wacks, D. (2016) How Christian was Iberia in the Middle Ages? And how can we tell? https://davidwacks.uoregon.edu/2016/04/14/pagan/
Wacks, D. (2017) Fairies and Pagan Mythologies in the Medieval Spanish Ballad. https://davidwacks.uoregon.edu/2017/12/31/fairies/
Whitehead (2018) Touching, crafting, knowing: religious artefacts and the fetish within animism, Body and Religion 2.2 224-244
Wikipedia (2020) List of Basque Mythological Figures https://en.wikipedia.org/wiki/List_of_Basque_mythological_figures#:~:text=Deities.%201%20Aide,%20a%20minor%20goddess%20of%20wind,normally%20imagined%20as%20a%20dragon%20or%20serpent.
Author Response
Thank you very much for your detailed review and the references, many of which I used as sources.
They helped me a lot to improve the manuscript.
I rewrote Chapter 3 in order to provide a better background to the history and the meaning of the Mary cult before analyzing the Spanish Marian sanctuaries.
Since I didn't use interviews (but mainly secondary sources), it was hard to interpret the exact way devotees interpret today the important "contractual relation" that seems to take place in these sanctuaries. Yet, I did add some extra references (like the one on on place attachment). This could very well be a future research question.
And yes, instead of "channeling"/"worship"/"holy"... "transmit"/"venerate"/"enspirited" are more precise terms
I hope to have addressed all your requests.
Thank you again for your valuable time.
Reviewer 3 Report
No comments for the resons I explain to the editors
Author Response
Thank you very much for your time and interest in this piece
Reviewer 4 Report
The article explores sacred sites in Spain, and vernacular religion. These sites have been studied before, and the author tries to bring an analysis which would incorporate vast variety of these sites. However, in the current form it is unclear how much the article brings as original input, since the author does not state this explicitly. Stating originality could help the reader to evaluate the scientific importance of this work. In my estimate it brings important ideas, but their originality and importance should be highlighted better. Moreover, the article does not state clearly which field of research this study belongs to.
Chapter 2 could elaborate the author's own research, investigation, more, what they did, in order to distinguish their work from previous research. Some of it has been explained, but a reader might seek more clarity in this.
Chapter 3 introduces Sanctuaries and previous research and knowledge. adequately. The chapter also discusses complexity and fluidity of the sites for people with differing motives (tourism-religion), albeit shortly. This discussion could be extended, or the author could add few other references concerning this topic.
Chapters 3 and 4 (and its subchapters) open with long quotes and short narratives, which could be incorporated better to the text, if wanted. A reader might see them as separate pieces - or not.
The chapter 4 discusses well with previous research and theories, adding the contribution of this article. It presents and explains the term geopiety, which I find useful in this case. A possible addition might be to discuss how this term relates to other similar discussions and historical trends, such as the search of romantic sublime in the nature in 1700s and 1800s (e.g. Edmund Burke).
On page 12 the phrase "key elements that articulate sacredness" needs clarification, how they articulate and why and to whom. It could be read as essentialistic claim, if not explained.
Concluding chapter 5 should begin with a concluding statement concerning the article. The current beginning can explain the starting conclusion after it. The question in the end of the first paragraph is unnecessary - the author should just tell what their conclusions are. The citation to Kraft and discussion of sacred-making should be mentioned in the text before the concluding chapter. It would make the article better, if this topic is introduced, for example, in chapter 3, and later incorporated into the text. The conclusions are good, but some parts could be introduced more explicitly in the text before the concluding chapter 5. The author should check that the conclusions and text form a perfect logical order of study.
In sum, the article presents a good case, but can be made better, and should be improved. Author's own input and the originality of the article should be made more clear, and the author should consider the comments above.
Author Response
Thank you for your time and interest in this piece of research
Yes, since there are very few academic papers on Spanish SNS, I tried to show the originality of the article more clearly, both in the introduction and in the discussions/conclusion chapter. I also specified the interdisciplinary character of the research (see Chapter 2 on research methods)
I added a chapter (now Chapter 3) and references in order to clarify the pilgrim-tourist debate and, hopefully, better contextualize the motivations of visitors to SNS.
Long quotes and short narratives were incorporated to the text.
Although the connection is interesting, I finally decided not to debate the "romantic sublime" relation to geopiey. The paper is already quite long.
I also expanded the Discussion/Conclusions chapter and stated more clearly the findings. The citation to Kraft and the discussion of sacred-making was introduced earlier (thank you, this will surely help the reader follow the argument).